# Forging Image Watermarks by Reversing Watermark Removal Attacks

## Abstract

Image generative models have accelerated the need for robust image watermarking to track and verify AI-generated images. While watermark removal attacks have been extensively studied, the threat of watermark forgery, where benign images are maliciously modified to appear watermarked, remains underexplored, especially in the no-box setting. In this work, we introduce WForge, a no-box and query-free forgery attack that reframes forgery as the inverse of removal. Our key insight is that residual perturbations from removal attacks approximate watermark signals and can be repurposed to forge watermarks. Concretely, we train a forger network to learn the pattern of residuals and apply it to unwatermarked images, making them falsely detected as watermarked. We evaluate WForge across three datasets and four state-of-the-art watermarking methods, demonstrating that it consistently outperforms existing forgery baselines. Our results further reveal a critical vulnerability: the existence of a successful removal attack implies the feasibility of forgery for the same watermarking method.

## 1 Introduction

Image generative models (Batifol et al., 2025; Google, 2025; Rombach et al., 2022) are now capable of producing high-quality, realistic images. While these models greatly expand the diversity of image content, they also raise ethical concerns and potential negative societal impacts, such as the spread of disinformation and copyright infringement (Dhaliwal, 2023; Escalante-De Mattei, 2023).

To address these concerns, image watermarking has been proposed and widely adopted as a proactive approach for tracking AI-generated images. For example, major companies such as OpenAI, Google, and Meta have already deployed watermarking services for their generative models (Gowal & Kohli, 2023; Mehdi, 2023). In addition, recent research (Jiang et al., 2024) has extended image watermarking techniques to support attribution of AI-generated content, further broadening their applicability. However, the security of existing watermarking methods has not been comprehensively analyzed—particularly against the challenge of *watermark forgery attacks*.

In the forgery attack, an adversary strategically modifies an image so that it is falsely recognized as watermarked by the detector. Such forged images can be exploited to undermine copyright attribution or to damage the reputation of watermarking service providers. Prior works have investigated the robustness of existing watermarking schemes against forgery. For instance, some studies (Yang et al., 2024a; Müller et al., 2025) have shown that simple averaging or DDIM inversion (Mokady et al., 2023) can effectively forge or remove watermarks, but these attacks are limited to specific methods such as Tree-Ring (Wen et al., 2023) and Gaussian Shading (Yang et al., 2024b). Other approaches (Wang et al., 2021) require access to paired watermarked and unwatermarked clean images, which is often impractical. In summary, existing forgery attacks are either not broadly applicable across diverse watermarking methods or rely on unrealistic assumptions about available resources.

To bridge this gap, we propose WForge, a no-box, query-free image watermark forgery attack. The key insight is to approximate watermark forgery as the inverse of watermark removal. Given a set of watermarked images, we first apply removal attacks to obtain watermark-removed versions, then compute the differences between the two, which we denote as *residuals*. We show that watermark patterns can be inferred and learned from these residuals. Concretely, we train a *forger network* via supervised learning: the network takes an image as input and predicts a residual that, when added to the image, causes it to be detected as watermarked.

We evaluate WForge across three datasets and multiple state-of-the-art image watermarking methods. Our results demonstrate that: (1) existing watermark forgery attacks are largely ineffective against state-of-the-art image watermarking methods; (2) WForge consistently outperforms forgery baselines across all three datasets, achieving high forgery success rates; and (3) the existence of a successful removal attack for an image watermarking method implies that the watermark can also be forged correspondingly. These findings reveal critical vulnerabilities in current image watermarking approaches and highlight the urgent need for more robust watermarking systems. Our main contributions are as follows:

- **Residual-based perspective.** We introduce a unified residual-based perspective that conceptually links watermark forgery and removal, revealing their intrinsic relationship.
- **Reversing removal for forgery.** Building on this perspective, we develop WForge and empirically demonstrate that reversing removal attacks into forgery provides a feasible and effective attack pathway, with forgery strength closely correlated with removal capability.
- **Comprehensive empirical evaluation.** We evaluate WForge against multiple state-of-the-art image watermarking methods across three datasets. WForge consistently outperforms baselines and is powerful enough to forge even localized watermarks.

## 2 RELATED WORK

### 2.1 IMAGE WATERMARKING

Image watermarks have been widely deployed as a proactive solution for AI-generated image provenance. Specifically, a watermarking method either embeds a post-hoc watermark into a given AI-generated image (Lu et al., 2025; Sander et al., 2025) or guides the generative model to produce an image with the watermark inherently embedded (Gunn et al., 2025). A watermark decoder (denoted as $D$) is then used to detect whether an image contains a watermark. Depending on whether the embedded watermark patterns are agnostic to or dependent on the image content, existing watermarking approaches can be categorized into two groups: content-agnostic and content-dependent.

**Content-agnostic methods**: This group of watermarking methods introduce fixed, content-agnostic patterns. For example, Tree-Ring (Wen et al., 2023) embeds bits into a ring-shaped Fourier region of the initial noise, while PRC watermark (Gunn et al., 2025) selects initial latents via pseudorandom error-correcting codes to achieve undetectability and stronger robustness. LaWa (Rezaei et al., 2024) adds structured residual signals in latent space, and SleeperMark (Wang et al., 2025) encodes messages as a trigger-based backdoor in the denoising process. However, these content-agnostic patterns leads to statistical bias, making their watermarks vulnerable to forgery attacks through statistical analysis such as simple averaging (Yang et al., 2024a).

**Content-dependent methods**: This group of methods embeds watermarks that depend on the image content. Specifically, the difference between a watermarked image and its unwatermarked version does not have statistical patterns. Representative approaches include RivaGAN (Zhang et al., 2019), which employs an attention-based embedding mechanism; StegaStamp (Tancik et al., 2020) employs an encoder–decoder framework jointly trained with differentiable perturbations, enabling robust decoding under real-world distortions; Stable Signature (Fernandez et al., 2023) refines the latent decoder of the image generator by conditioning it on a binary signature; and the more recent Watermark Anything Model (WAM) (Sander et al., 2025), which enables localized watermarking by imperceptibly modifying selected regions of an image and later extracting the watermark from those regions. Because these methods depend on the content, their watermarks resist simple averaging or statistical analysis. Our experiments further demonstrate that current forgery attacks are ineffective against them. In this work, we propose WForge to address this group of image watermarks.

### 2.2 WATERMARK REMOVAL ATTACKS

This work is designed on the idea of reversing watermark removal and thus relies on watermark removal attacks (Jiang et al., 2023; Zhao et al., 2024; Hu et al., 2025; Kassis & Hengartner, 2025; Liu et al., 2025), which are used to remove the watermark from watermarked images without sacrificing visual quality. For instance, Zhao et al. (2024) proposed regeneration methods that add noise to disrupt watermarks and reconstruct images with generative models such as Variational Auto-Encoders

(VAEs) (Kingma & Welling, 2013) or diffusion models (Ho et al., 2020). Later, Liu et al. (2025) advanced this approach with a controllable diffusion method. Jiang et al. (2023) proposed adding imperceptible perturbations to evade watermark detection, but WEvade requires white-box or black-box access, and thus is not applicable in our no-box scenarios.

### 2.3 WATERMARK FORGERY ATTACKS

Watermark forgery attacks (Wang et al., 2021; Li et al., 2023; Yang et al., 2024a; Müller et al., 2025) aim to forge a watermark for an unwatermarked image by adding a human-imperceptible perturbation. For example, Watermark Faker (Wang et al., 2021) trains a U-Net to transform an unwatermarked image into a watermarked one using paired watermarked and unwatermarked images. However, clean images may be inaccessible in practice. To address this, we adapt the method by treating watermark-removed images as unwatermarked images. Watermark Steganalysis (Yang et al., 2024a) demonstrates that simple averaging can successfully forge or remove the watermark for content-agnostic methods like Tree-Ring and Gaussian Shading. However, our experiments show that these methods are ineffective against content-dependent watermarking methods.

## 3 PROBLEM DEFINITION

**Image watermark forgery**: The forgery attack aims to forge the watermark for a non-watermarked images via perturbing it, i.e., misleads the watermark detector to falsely detect the perturbed image as watermarked. At the same time, the visual quality should be preserved as well as possible in the perturbed image. Formally, we define the problem as follows:

**Definition 1** (Image Watermark Forgery). Given a non-watermarked image $I_u$ and the watermark detector $D$, the forgery attack aims to find a perturbation $\delta_{\text{forge}}$ that satisfies:

$$D(I_u + \delta_{\text{forge}}) = \neg D(I_u), \qquad \|\delta_{\text{forge}}\| \leq \epsilon_1, \tag{1}$$

where $D$ can be treated as a binary classifier, i.e., $D(x) = \mathbb{I}(\text{Image } x \text{ is detected as watermarked})$, $\mathbb{I}$ is the indicator function, $I_u + \delta_{\text{forge}}$ is the perturbed image, $\neg$ means reversing the classification result, and $\epsilon_1$ is a threshold for the perturbation $\delta_{\text{forge}}$ to ensure the image's visual quality is well-preserved.

**Image watermark removal**: The removal attack aims to perturb a watermarked image such that the watermark detector falsely detect the perturbed image as non-watermarked, while preserving the visual quality. Formally, we define it as follows:

**Definition 2** (Image Watermark Removal). Given a watermarked image $I_w$ and the watermark detector $D$, the removal attack aims to find a perturbation $\delta_{\text{remove}}$ that satisfies:

$$D(I_w + \delta_{\text{remove}}) = \neg D(I_w), \qquad \|\delta_{\text{remove}}\| \leq \epsilon_2, \tag{2}$$

where $I_w + \delta_{\text{remove}}$ is the perturbed image and $\epsilon_2$ ensures the image's visual quality is well-preserved.

### 3.1 THREAT MODEL

**Attack's goal**: Given an unwatermarked image $I_u$, the attacker aims to construct a perturbed image $I_u + \delta_{\text{forge}}$ such that (i) $I_u + \delta_{\text{forge}}$ is visually indistinguishable from $I_u$ under human perception (e.g., $\|\delta_{\text{forge}}\| \leq \epsilon_1$ and high perceptual similarity), and (ii) $I_u + \delta_{\text{forge}}$ is detected as *watermarked* by the watermark detector. A successful attack therefore produces a forged watermark that can be misused as false proof of ownership or authentication, undermining copyright attribution, content integrity verification, and the credibility of the watermarking system.

**Attack's capability**: We assume a *no-box* setting. In such a setting, the access to the watermark detector $D$ and even its API is prohibited, which means the attacker here cannot interact with the watermarking system at all. Instead, the only available resource is a collection of $N$ watermarked images generated by the target watermarking method, without corresponding unwatermarked counterparts. This captures practical scenarios where leaked or publicly available watermarked content is accessible, but the watermarking system itself is not.

**Attacker's background knowledge**: The attacker has no knowledge of the watermarking algorithm, parameters, or design principles. The only knowledge is that the accessible images have been watermarked using the same watermarking scheme. The attacker may attempt to infer useful clues from these watermarked images, but no further information is assumed.

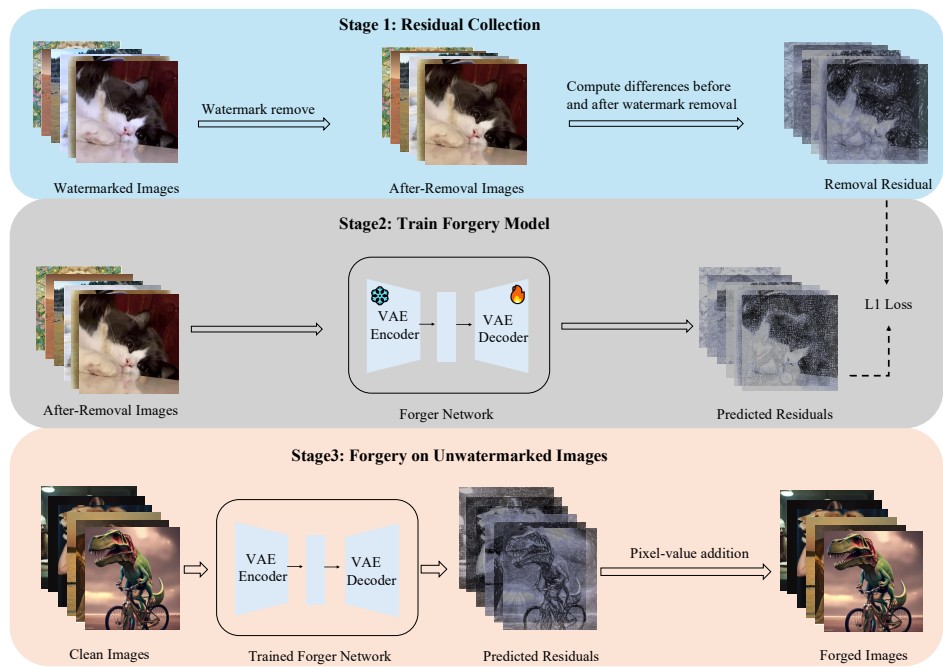

Figure 1: The overall framework of WForge. The pipeline consists of three stages: (1) residual collection, where watermark-removal outputs are compared with the original watermarked images to obtain removal residuals; (2) training the forgery model to learn and predict such residuals; and (3) applying the trained forger to clean images, adding predicted residuals to generate forged images.

## 4 METHODOLOGY

**Overview of our WForge**: When the attacker has no knowledge of the watermarking algorithm, the core challenge of a forgery attack is constructing an alternative embedding mechanism that produces a perturbation closely matching the genuine watermark. In this work, we propose WForge, a no-box, query-free forgery method for content-dependent image watermarks. The framework diagram is depicted in Figure 1, and the detailed procedure is provided in Algorithm 1. WForge is built on three key observations: (1) watermarks produced by content-dependent methods exhibit patterns that strongly depend on the input image; (2) a successful watermark removal attack actually removes these patterns by introducing a perturbation; and (3) the removal perturbation can be recovered and repurposed, by reversing the removal attack, to forge the watermark. Concretely, WForge requires a collection of watermarked images. We apply an effective watermark removal attack to these images to obtain their watermark-removed versions and compute the differences (which we call *residuals*). We then train a forger network in a supervised way to predict residuals: it takes an image as input and outputs a predicted residual. After learning from these residuals, the trained forger can predict a residual for an unseen clean image; when this residual is added to the image, the watermark detector could extract a watermark, enabling successful forgery.

### 4.1 RETHINKING WATERMARK FORGERY AND WATERMARK REMOVAL

**A residual-based perspective**: We can view the watermark embedding process from a residual-based perspective. Let $I_u$ denote an unwatermarked image, $I_w$ its watermarked version, and $E$ the watermark embedding algorithm. The embedding process can then be formulated as:

$$I_w = E(I_u) = I_u + r(I_u), \tag{3}$$

where $r(I_u)$ denotes a small, image-dependent perturbation that encodes the watermark. Definitions 1 and 2 can be interpreted as inverse formulations, in which the perturbation $r(I_u)$ simulta-

neously satisfies both objectives: forging a watermark in $I_u$ and removing the watermark from $I_w$. Accordingly, watermark forgery and removal can be seen as finding perturbations $\delta_{\text{forge}}(I_u) \approx r(I_u)$ and $\delta_{\text{remove}}(I_w) \approx -r(I_u)$, respectively. This observation suggests that successful forgery and removal perturbations are approximately inverse, i.e.,

$$\delta_{\text{forge}}(I_u) \approx r(I_u) \approx -\delta_{\text{remove}}(I_w). \tag{4}$$

In other words, watermark *forgery* and *removal* can be regarded as approximately inverse problems governed by the same residual. We formally define this reversibility as follows:

*Lemma* 1 (Reversibility of Watermark Forgery and Removal). Assume there exists an ideal watermark removal method $\mathcal{R}$ such that, for any watermarked image $I_w$, $\mathcal{R}$ perfectly removes the watermark and exactly recovers the unwatermarked image $I_u$, i.e., $\mathcal{R}(I_w) = I_u$. In this case, the corresponding removal perturbation is $\delta_{\text{remove}}(I_w) = \mathcal{R}(I_w) - I_w = -r(I_u)$. Further, assume that $\mathcal{R}$ can be ideally reversed into a forgery method $\mathcal{F}$, such that $\mathcal{F}(I_u) = \delta_{\text{forge}}(I_u) = -\delta_{\text{remove}}(I_w)$. Then the forgery method $\mathcal{F}$ is guaranteed to successfully forge the watermark for every unwatermarked image $I_u$, that is, $I_u + \delta_{\text{forge}}(I_u) = I_w$.

**From ideal assumptions to practical solutions**: Lemma 1 relies on two following assumptions.

1. **Assumption for the ideal removal method $\mathcal{R}$.** In practice, no removal method can perfectly remove the watermark by exactly recovering $I_u$. Our *practical solution* is to employ a strong removal method $\mathcal{R}'$, which removes the watermark from $I_w$ by introducing a perturbation $\delta_{\text{remove}}$ that closely approximates $-r(I_u)$, i.e.,

$$\delta_{\text{remove}}(I_w) = \mathcal{R}'(I_w) - I_w \approx -r(I_u). \tag{5}$$

   We define this perturbation $\delta_{\text{remove}}(I_w)$ as the *ground-truth residual*, denoted by $R_g$.

2. **Assumption for the exact inversion of the removal method $\mathcal{R}$.** In practice, the forgery task is not simply about recovering the same residual from the same image. Instead, the attacker requires a method that generalizes across unseen unwatermarked images. Our *practical solution* is to approximate this inversion process in a data-driven manner: we collect a set of image–residual pairs $\{(\mathcal{R}'(I_w), R_g)\}$ obtained from the watermark removal process, and train a model $M_\theta$ in a supervised way to learn the mapping from the watermark-removed image $\mathcal{R}'(I_w)$ to a predicted residual $R_p$. Formally,

$$R_p = M_\theta(\mathcal{R}'(I_w)) \approx R_g. \tag{6}$$

   Here, $M_\theta$ is not expected to exactly invert the removal method $\mathcal{R}$ for each individual pair, but rather to generalize so that $R_p$ can act as a surrogate watermark, enabling successful forgery of arbitrary unwatermarked images $I_u$.

In what follows, Section 4.2 introduces how residuals are obtained via watermark removal methods to provide supervision. Section 4.3 then describes how to learn residuals with the model $M_\theta$ such that $I_u + M_\theta(I_u)$ is detected as watermarked. Finally, Section 4.4 presents empirical evidence supporting our three key observations.

## 4.2 OBTAINING RESIDUALS

We consider an attacker who has access only to a collection of watermarked images $\{I_w^{(i)}\}_{i=1}^N$. From the residual perspective, the embedded watermark is modeled as a small, image-dependent perturbation. To obtain supervision for training, we derive ground-truth residuals using the watermark removal method $\mathcal{R}'$. Specifically, given a watermarked image $I_w^{(i)}$, the method outputs a watermark-removed version $\mathcal{R}'(I_w^{(i)})$. The corresponding ground-truth residual $R_g^{(i)}$ is then computed as $\mathcal{R}'(I_w^{(i)}) - I_w^{(i)}$. Finally, to preserve visual quality, we project the residual onto the admissible norm ball via $\Pi_{\epsilon_2}$:

$$R_g^{(i)} \leftarrow \Pi_{\epsilon_2}\big(R_g^{(i)}\big), \text{ such that } \|R_g\| \leq \epsilon_2. \tag{7}$$

Intuitively, $\mathcal{R}'$ removes the watermark while maintaining the image's visual quality. Consequently, the residual $R_g^{(i)}$ serves as a surrogate for the watermark. The set $\{(\mathcal{R}'(I_w^{(i)}), R_g^{(i)})\}_{i=1}^N$ is collected as the training dataset, providing supervision for learning a forger network $M_\theta$ that generalizes to unseen unwatermarked images.

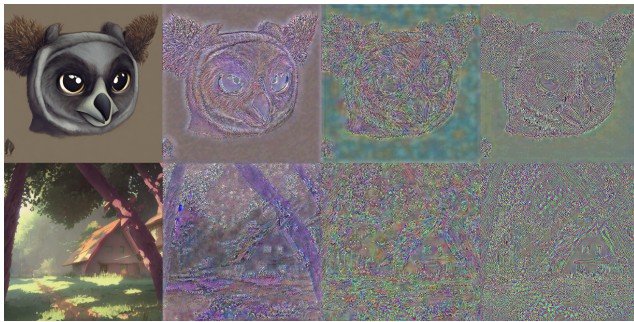

Figure 2: Visualization of original images and three types of residuals (scaled by $\times 10$) in the Stable Signature watermark. Column 1: unwatermarked images $I_u$; Column 2: watermark residuals $r(I_u) = I_w - I_u$; Column 3: residuals $R_g = \mathcal{R}'(I_w) - I_w$ obtained via a VAE-based remover $\mathcal{R}'$; Column 4: our predicted residuals $R_p = M_\theta(I_u)$. Additional visual results are provided in Figure 11 of the appendix.

## 4.3 LEARNING RESIDUALS

Following Section 4.2, we construct a training dataset $\mathcal{T} = \{(\mathcal{R}'(I_w^{(i)}), R_g^{(i)})\}_{i=1}^N$, where $\mathcal{R}'(I_w^{(i)})$ denotes the watermark-removed version of $I_w^{(i)}$ and $R_g^{(i)}$ is the corresponding ground-truth residual. We train a forger network $M_\theta$ to predict a residual given an image $I$. Formally,

$$I_f = I + \Pi_{\epsilon_1}\big(M_\theta(I)\big), \tag{8}$$

where $I_f$ denotes the watermark-forged version of image $I$, which is likely to be detected as watermarked by the detector $D$ since the predicted residual $R_p = M_\theta(I)$ serves as a surrogate watermark. To preserve visual quality, we apply the projection $\Pi_{\epsilon_1}$. To capture the hidden watermark pattern from residuals, we formulate the following optimization problem and solve it via gradient descent:

$$\min_\theta \sum_{i=1}^N \big\|\Pi_{\epsilon_1}\big(M_\theta(\mathcal{R}'(I_w^{(i)}))\big) - R_g^{(i)}\big\|_1, \tag{9}$$

where $\|\cdot\|_1$ denotes the $\ell_1$ norm.

## 4.4 VALIDATING OUR OBSERVATIONS

We empirically validate three key observations underpinning our approach: (i) watermark residuals are image-dependent—they carry structure correlated with the input image; (ii) a successful watermark removal attack could remove watermark patterns, so that removal perturbations $I_w - \mathcal{R}'(I_w)$ closely match the ground-truth watermark residuals $r(I_u) = I_w - I_u$; and (iii) these removal perturbations can be recovered and repurposed, by reversing the removal attack, to forge the watermark. Accordingly, the predicted residual for $I_u$ produced by our forgery method $R_p = M_\theta(\mathcal{R}'(I_u))$ should closely resemble the removal perturbation, and consequently also align with the ground-truth watermark residuals $r(I_u)$.

**Obs. 1: Watermarks are dependent on images**: The first two columns of Figure 2 demonstrate that watermarks concentrate along edges and textures, reflecting their dependence on image content.

**Obs. 2: Residuals approximate the watermarks**: Figure 2 shows that Column 3 ($R_g$) resembles Column 2 ($r(I_u)$), indicating that the residuals introduced by the remover $\mathcal{R}'$ serve as good surrogates for the watermarks.

**Obs. 3: Predicted residuals approximate ground-truth residuals**: As shown in Columns 3 and 4 of Figure 2, the predicted residuals $R_p$ both visually and statistically resemble the ground-truth residuals $R_g$ introduced by the remover $\mathcal{R}'$, and thus align closely with $r(I_u)$.

# 5 EVALUATION

## 5.1 EXPERIMENTAL SETUP

**Datasets**: Our training dataset consists of 5,000 watermarked images generated from the MS-COCO (Lin et al., 2014) training set. For evaluation, we use three testing datasets: the MS-COCO testing set, ImageNet (Deng et al., 2009), and DiffusionDB (Wang et al., 2022). From each testing dataset, we randomly sample 1,000 unwatermarked images. All images are resized to $512 \times 512$ to ensure comparability.

**Image watermarking methods**: We evaluate four state-of-the-art image watermarking methods: RivaGAN (Zhang et al., 2019), StegaStamp (Tancik et al., 2020), Stable Signature (Fernandez et al., 2023), and Watermark Anything Model (WAM) (Sander et al., 2025). RivaGAN, StegaStamp, and WAM belong to the category of post-hoc watermarking methods, which embed a watermark into an image after it has been generated. For these methods, we generate watermarked images by applying the method to original unwatermarked images randomly sampled from MS-COCO. In contrast, Stable Signature represents model-integrated watermarking methods, where the watermarking process is incorporated directly into the image generative model so that all generated images are inherently watermarked. Unlike the other three methods, the training dataset of Stable Signature consists of watermarked images generated from 5,000 captions randomly sampled from MS-COCO.

**Baseline methods**: We evaluate two forgery baselines: Watermark Steganalysis (Yang et al., 2024a) and Watermark Faker (Wang et al., 2021). All forgery attacks are considered under a no-box setting, where the attacker has access only to watermarked images and the corresponding original unwatermarked images are unavailable.

**Evaluation metrics**: We evaluate each forgery method along two dimensions: Effectiveness and Utility. A forged image is considered successful if the average bitwise accuracy between the watermark extracted by the watermarking method and the ground-truth watermark exceeds a threshold $\tau$. For Effectiveness, we report two metrics: *AvgBitAcc*, the average bitwise accuracy across all testing images; and *Success Rate*, the proportion of forged images whose bitwise accuracy is above the fixed threshold $\tau = 0.8$. For Utility, we assess the visual quality of forged images relative to their non-watermarked counterparts using PSNR and FID (Heusel et al., 2017).

**Implementation details**: Comprehensive details, including hyperparameters, training settings, and baseline implementations, are provided in Appendix A.2.

## 5.2 MAIN RESULTS

Table 1: Forgery results on MS-COCO, ImageNet, and DiffusionDB.

| Dataset | Method | Watermark Steganalysis | | | | Watermark Faker | | | | WForge | | | |
|---|---|---|---|---|---|---|---|---|---|---|---|---|---|
| | | AvgBitAcc↑ | Success Rate↑ | PSNR↑ | FID↓ | AvgBitAcc↑ | Success Rate↑ | PSNR↑ | FID↓ | AvgBitAcc↑ | Success Rate↑ | PSNR↑ | FID↓ |
| MS-COCO | RivaGAN | 0.527 | 0.000 | 34.88 | 1.596 | 0.501 | 0.000 | 20.25 | 55.43 | 0.968 | 0.967 | 34.39 | 8.760 |
| | StegaStamp | 1.000 | 1.000 | 32.27 | 6.483 | 0.490 | 0.000 | 19.78 | 60.45 | 0.874 | 0.968 | 32.71 | 11.31 |
| | Stable Signature | 0.464 | 0.000 | 30.65 | 2.812 | 0.539 | 0.000 | 19.86 | 58.24 | 0.955 | 0.975 | 32.15 | 11.41 |
| | WAM | 0.464 | 0.000 | 35.01 | 2.495 | 0.477 | 0.000 | 20.18 | 62.16 | 0.952 | 0.994 | 31.38 | 13.13 |
| ImageNet | RivaGAN | 0.526 | 0.000 | 34.88 | 1.187 | 0.492 | 0.000 | 20.46 | 44.93 | 0.951 | 0.924 | 35.44 | 6.126 |
| | StegaStamp | 1.000 | 1.000 | 32.28 | 4.467 | 0.492 | 0.000 | 19.90 | 47.38 | 0.875 | 0.963 | 33.08 | 7.996 |
| | Stable Signature | 0.463 | 0.000 | 30.69 | 2.060 | 0.531 | 0.000 | 19.96 | 47.67 | 0.935 | 0.955 | 32.70 | 8.198 |
| | WAM | 0.466 | 0.000 | 35.02 | 1.651 | 0.473 | 0.000 | 20.29 | 50.16 | 0.920 | 0.863 | 32.54 | 9.900 |
| DiffusionDB | RivaGAN | 0.512 | 0.000 | 34.90 | 1.636 | 0.481 | 0.000 | 20.39 | 55.63 | 0.891 | 0.793 | 32.57 | 12.13 |
| | StegaStamp | 1.000 | 1.000 | 32.30 | 7.354 | 0.491 | 0.000 | 19.55 | 63.10 | 0.873 | 0.926 | 30.84 | 17.23 |
| | Stable Signature | 0.473 | 0.000 | 30.69 | 2.915 | 0.533 | 0.000 | 19.91 | 61.68 | 0.931 | 0.957 | 31.73 | 13.60 |
| | WAM | 0.469 | 0.000 | 35.03 | 2.488 | 0.478 | 0.000 | 20.16 | 61.24 | 0.876 | 0.764 | 29.76 | 19.59 |

Table 1 shows that WForge consistently achieves high average bitwise accuracy in forgery attacks across four watermarking methods and three datasets—typically above 90%, with the lowest at 87%. The Watermark Steganalysis baseline, although preserving satisfactory image quality, is completely ineffective against RivaGAN, Stable Signature, and WAM. The Watermark Faker baseline also fails to forge watermarks and further introduces substantial quality degradation. In contrast, WForge substantially outperforms both baselines, attaining consistently high success rates across different watermarking methods and datasets while maintaining visual quality. Figure 3 further reports success rates under varying detection thresholds $\tau$, where Success Rate@$\tau$ denotes the success rate when the threshold is set to $\tau$. Results indicate that increasing $\tau$ gradually decreases success rates; nevertheless, WForge maintains strong performance even at $\tau = 0.9$.

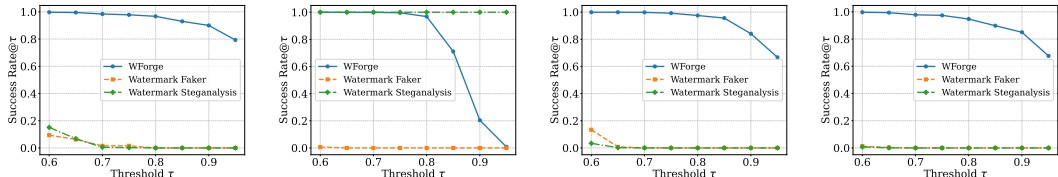

Figure 3: Success Rate@$\tau$ results of two baselines and WForge against different watermarking methods on MS-COCO. From left to right: RivaGAN, StegaStamp, Stable Signature, and WAM.

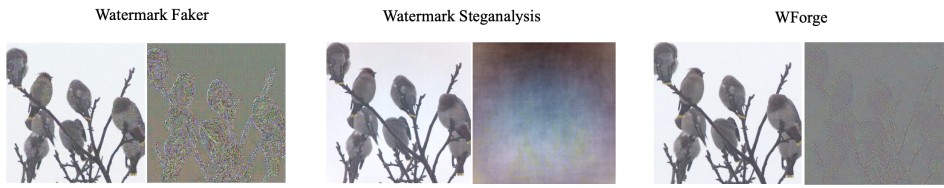

Figure 4: Visual comparison of forged images and their corresponding predicted residuals (magnified ×10) across three forgery methods against Stable Signature.

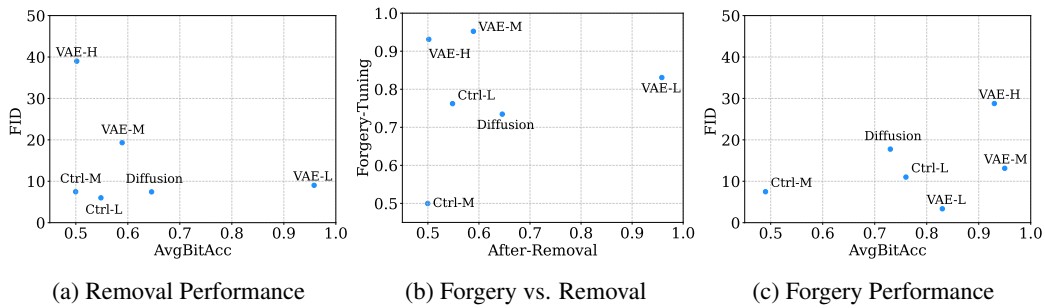

| (a) Removal Performance | (b) Forgery vs. Removal | (c) Forgery Performance |
|---|---|---|

Figure 5: Comparisons with different removal methods. The first subfigure reports AvgBitAcc (on watermark-removed images) and FID results. The second subfigure analyzes the relationship between removal and forgery in terms of AvgBitAcc. The third subfigure reports AvgBitAcc (on watermark-forged images) and FID results.

To further illustrate the underlying differences, Figure 4 visualizes the perturbations produced by three forgery methods and added to the image during forgery. The perturbation from Watermark Steganalysis does not capture individual image characteristics, as it is obtained by simple averaging across a group of images. Watermark Faker generates a perturbation that depends on image content but contains obvious artifacts. In contrast, WForge introduces a much smaller, human-imperceptible perturbation. These results demonstrate that WForge is more effective and preserves visual quality better than the two baselines.

## 5.3 ABLATION STUDY

To evaluate the impact of parameter choices, we performed a series of ablation studies. Unless otherwise specified, all configurations follow the default settings, with only the parameter under investigation varied. Experiments are conducted using WAM on the MS-COCO dataset.

**WForge using different removal methods**: We conduct experiments with three watermark removal methods: VAE regeneration (VAE) (Zhao et al., 2024), Diffusion regeneration (Diffusion) (Zhao et al., 2024), and CtrlRegen (Ctrl) (Liu et al., 2025). There are three strength levels (Low, Medium, and High), corresponding to the magnitude of the introduced perturbations. We use notations such as VAE-M to denote VAE regeneration at medium strength and Ctrl-L to denote CtrlRegen at low strength. A complete comparison across all methods is provided in Table 4 in the Appendix.

Figure 5c shows that forgery performance varies across removal methods, even when their watermark removal performance is comparable. For example, as illustrated in Figure 5b, VAE-M, Ctrl-L,

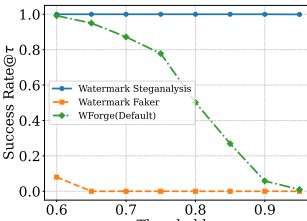 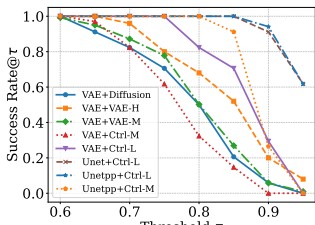 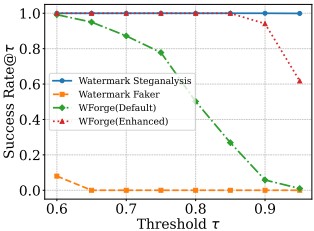

Figure 6: The left subfigure shows the comparison of our method with other baselines on LaWa under the default setting (VAE+VAE-M). The middle subfigure presents the forgery performance on LaWa with different combinations. The right subfigure illustrates the results obtained by selecting better-performing configurations for LaWa.

and Diffusion all achieve similar post-removal *AvgBitAcc* (measured on watermark-removed images) of about 0.6. However, VAE-M produces the strongest forgery, Ctrl-L performs moderately, and Diffusion performs the worst. This indicates that different removal methods eliminate watermarks in distinct ways. As a result, the forger network learns different watermark patterns, some effective and others not, leading to the observed disparities in forgery performance. To further highlight these differences, Figure 9 in the Appendix visualizes the distinct residual distributions when different removal methods are used.

Results reveal a conditional positive correlation between watermark removal and forgery: when the removal method preserves visual quality, stronger removal performance leads to more successful forgery. In Figure 5b, a direct comparison between VAE-L and VAE-M shows that the model with stronger removal achieves higher forgery performance. However, once removal substantially degrades image quality, this relationship no longer holds. For example, VAE-H (FID = 38.99) in Figure 5a injects excessive perturbations, rendering the residuals unusable for learning and thereby degrading forgery performance, as shown in Figure 5c.

**WForge using different forger networks**: We evaluate different backbone architectures for the forger network in residual learning, including UNet (Ronneberger et al., 2015), UNetpp (Zhou et al., 2018), FPN (Lin et al., 2017), and VAE (Rombach et al., 2022). Unless otherwise mentioned, we use pretrained versions of these models, freeze the encoder parameters, and fine-tune only the decoder. Based on the results in Table 2, we observe that different backbone architectures yield varying levels of forgery performance. For the WAM method, the UNet family achieves the highest forgery success rate while maintaining high image quality. In contrast, FPN performs the worst and entirely fails at the forgery task. For VAE-based models, fine-tuning the decoder achieves a higher success rate compared to fine-tuning the encoder, but this improvement comes at the cost of significantly degraded image quality.

Table 2: Forgery results using different forger networks

|  | AvgBitAcc↑ | Success Rate↑ | PSNR↑ | FID↓ |
|---|---|---|---|---|
| UNet | 0.968 | 0.964 | 42.76 | 3.645 |
| UNetpp | 0.964 | 0.960 | 40.53 | 6.599 |
| FPN | 0.505 | 0.000 | 49.38 | 0.922 |
| VAE FT-Encoder | 0.898 | 0.833 | 44.01 | 2.070 |
| VAE FT-Decoder | 0.952 | 0.947 | 31.38 | 13.13 |

**Different combinations on forgery effects**: The optimal forgery configuration for a given watermarking method is not fixed. This variability arises because different removal methods remove watermarks in distinct ways, while different network architectures learn residuals with varying effectiveness. Consequently, combining removal methods with network architectures produces diverse forgery performance. In Figure 6, we evaluate the LaWa watermarking method (Rezaei et al., 2024) under its default setting and identify its optimal forgery configuration by varying both components. We observe that the combination of UNetpp and Ctrl-L significantly outperforms the default configuration.

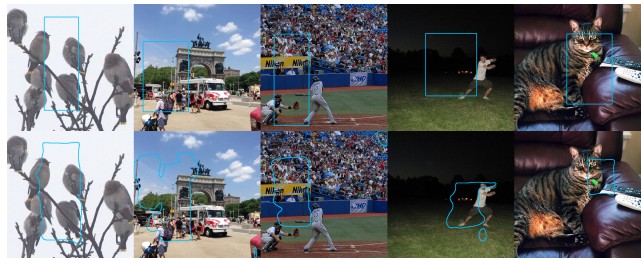

Figure 7: Attack against the localized watermark property. In the first row, we present the forged images where localized forgery residuals are embedded by randomly selecting 20% of the area in the cover images as a mask. The blue boxes indicate the regions where the localized forgery residuals were added. The second row presents the predicted mask locations obtained by the WAM decoder.

**WForge against localized watermarks**: WAM is a watermarking method in which the watermark is sparsely embedded within localized regions rather than across the entire image. Despite this design, WForge successfully learns a residual pattern that extends over the full image, achieving high forgery accuracy and effectively deceiving the watermark detector. Moreover, We show that our forged residuals undermine WAM's core property of resisting localized tampering, as shown in Figure 7. Additional visualizations and evaluations are provided in Appendix A.3.

**Other**: We also analyze the impact of training sample size on forgery performance. Detailed results are provided in Appendix A.4.2.

## 6 CONCLUSION AND FUTURE WORK

In this work, we introduce WForge, a no-box, query-free forgery attack against image watermarks. By framing watermark forgery and removal as approximately inverse problems, we demonstrated that residuals derived from removal can be effectively repurposed for forgery, achieving high success rates while preserving visual quality across diverse datasets and watermarking methods. A key insight from our study is that the existence of a successful removal attack for a watermarking method inherently implies the possibility of forging that watermark, exposing fundamental vulnerabilities in current state-of-the-art watermarking approaches. Future work includes extending WForge and the residual-based perspective to other modalities such as video and audio, and designing watermarking defenses that explicitly mitigate the reversibility between removal and forgery.

## 7 ETHICS STATEMENT

Our study reveals that WForge, a no-box, query-free, and highly effective forgery attack on image watermarks, poses potential risks of misuse, including the undermining of copyright claims, content provenance, and trust in watermarking systems. By reversing removal attacks, the method can be exploited to produce forged ownership evidence with practical feasibility, since it requires only access to a collection of watermarked images and no knowledge of the underlying watermarking algorithm. We emphasize that this work is conducted solely for research purposes, and that responsible development and deployment are essential. Ultimately, by uncovering these vulnerabilities, our findings aim to support the community in designing more robust and trustworthy watermarking methods.

## 8 REPRODUCIBILITY STATEMENT

Our main contributions are as follows: (i) we introduce a unified residual-based perspective that conceptually links watermark forgery and removal; (ii) building on this perspective, we develop WForge and show that reversing removal into forgery provides a feasible and effective attack pathway; and (iii) we evaluate WForge against multiple state-of-the-art image watermarking methods. The hypothesis, reasoning and observation underlying the unified residual-based perspective are elaborated in Section 4. We provide a clear description of the experimental setup in Section 5.1, while comprehensive details are presented in Appendix A.2 including hyperparameters, training

settings, watermarking methods, baseline implementations, and the configuration of WForge to ensure reproducibility. Our results can be reproduced using publicly available GitHub repositories, and the pretrained weights are available on Hugging Face.

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

# A APPENDIX

## A.1 USE OF LLMS

We use large language models to aid or polish writing at the sentence level, such as fixing grammar and re-wording sentences. LLMs were not involved in designing methods, conducting experiments, or drawing conclusions. No sensitive or proprietary data were shared with LLMs.

## A.2 IMPLEMENTATION DETAILS

### A.2.1 WATERMARKING METHODS

Table 3: Watermarking settings used in our experiments.

| Watermark scheme | Bit Length |
|---|---|
| Tree-Ring | \ |
| LaWa | 48 |
| SleeperMark | 48 |
| PRC watermark | 64 |
| RivaGAN | 32 |
| StegaStamp | 100 |
| Stable Signature | 48 |
| WAM | 32 |

For watermarking methods used in our experiments, we adopt the official implementations whenever possible. The embedding bit lengths for each method are summarized in Table 3. For WAM, the watermarking region is randomly restricted to 50% of the image by default.

### A.2.2 BASELINES

Müller et al. (2025) investigates watermarks embedded in the initial latent noise of diffusion models and their forgery process depends on DDIM inversion totally, which is not suitable for our comparison. Finally, we select the following two methods as our baselines.

For Watermark Steganalysis (Yang et al., 2024a), we collect 5,000 watermarked images and 5,000 out-of-distribution clean images from the Open Images dataset (Kuznetsova et al., 2020) to estimate and extract the average watermark residual, which is then used to generate forged images.

For Watermark Faker (Wang et al., 2021), whose original implementation requires paired original and watermarked images for training, For fairness, we replace the original images with watermark-removed images obtained using the same removal method as our approach. We conduct our experiments entirely based on the official code, with only minor modifications: the input image size is increased from 256 to 512, the number of training epochs is set to 40, and the batch size to 16.

### A.2.3 WFORGE

By default, WForge adopts the VAE (Ballé et al., 2018) from the CompressAI library (Bégaint et al., 2020) as the watermark remover, using the official hyperparameter settings from the removal method proposed by Zhao et al. (2024). For the residual learning network, we employ the pre-trained VAE from Stability AI (AI, 2022), freeze the encoder, and fine-tune the decoder.

We note that while our default settings allow successful forgery across the aforementioned watermarking schemes, they are not necessarily tuned to achieve the best performance for each specific method. Our goal is to demonstrate the feasibility of forging image watermarks by reversing watermark removal attacks, and in Section 5.3 we further show that different parameter combinations can yield varying effectiveness across different watermarking schemes.

WAM Watermark                                    WForge

Figure 8: Visual comparison of the forged image and ground-truth watermark image in WAM. Left: watermarked image, its watermark residual (scaled by $\times 10$), and WAM-predicted mask. Right: forged image with globally embedded residual, its residual (scaled by $\times 10$), and WAM-predicted mask.

## A.3 CASE STUDY: SUPPLEMENTARY EVALUATION OF WAM

In post-hoc watermarking, Watermark Anything (WAM) is particularly challenging because it resists localized tampering. Unlike conventional schemes embedding watermarks across the whole image, WAM inserts them only in selected regions, predicting the location before extraction. Thus, watermark signals appear sparsely, leaving large blank areas that complicate forgery attacks.

Even with partially watermarked training samples, our method effectively learns residual patterns that achieve high bitwise forgery accuracy and deceive the detector. As shown in Figure 8, when applied to the entire cover image, our forged residuals lead to distorted masks that still mislead the decoder. These results confirm that our method generalizes from partial to full-image cases, underscoring its robustness.

To further validate, we embedded forged residuals into part of a cover image. As shown in Figure 7, predicted watermark locations largely overlap with forged regions, confirming that our method undermines WAM's resistance to localized tampering. This result demonstrates that our method is capable of undermining the core property of WAM, namely its resistance to localized tampering.

## A.4 SUPPLEMENTARY OF ABLATION STUDY

### A.4.1 REMOVER

In our experiments, we systematically compared five methods (VAE-L/M/H, Ctrl-L/M, and Diffusion). However, Ctrl-H was excluded from the evaluation because the reconstruction-based removal in Ctrl-M had already introduced substantial degradation to the original image, rendering tuning to forgery ineffective. Consequently, extending the evaluation to Ctrl-H was neither feasible nor meaningful.

Table 4: Forgery results on different remove methods. After Removal presents the performance of each method in terms of watermark removal, evaluated by effectiveness and utility. Tuning Removal to Forgery shows the forgery performance after adapting the corresponding methods with WForge.

|  | After Removal | | | Tuning Removal to Forgery | | |
|---|---|---|---|---|---|---|
|  | AvgBitAcc↓ | PSNR↑ | FID↓ | AvgBitAcc↑ | PSNR↑ | FID↓ |
| VAE-L | 0.958 | 32.48 | 9.027 | 0.831 | 39.18 | 3.350 |
| VAE-M | 0.589 | 29.51 | 19.31 | 0.952 | 31.38 | 13.13 |
| VAE-H | 0.502 | 26.95 | 38.99 | 0.931 | 27.08 | 28.77 |
| Ctrl-L | 0.548 | 23.11 | 6.003 | 0.762 | 29.97 | 11.01 |
| Ctrl-M | 0.500 | 20.96 | 7.476 | 0.500 | 20.96 | 7.476 |
| Diffusion | 0.646 | 22.74 | 7.439 | 0.735 | 28.34 | 17.76 |

### A.4.2 TRAINING SIZE

We studied the effect of varying the number of training samples on the forgery performance. By gradually increasing the training set size, we observed how data scale influences the forgery performance.

| VAE-L | VAE-M | VAE-H | Ctrl-L | Ctrl-M | Diffusion |

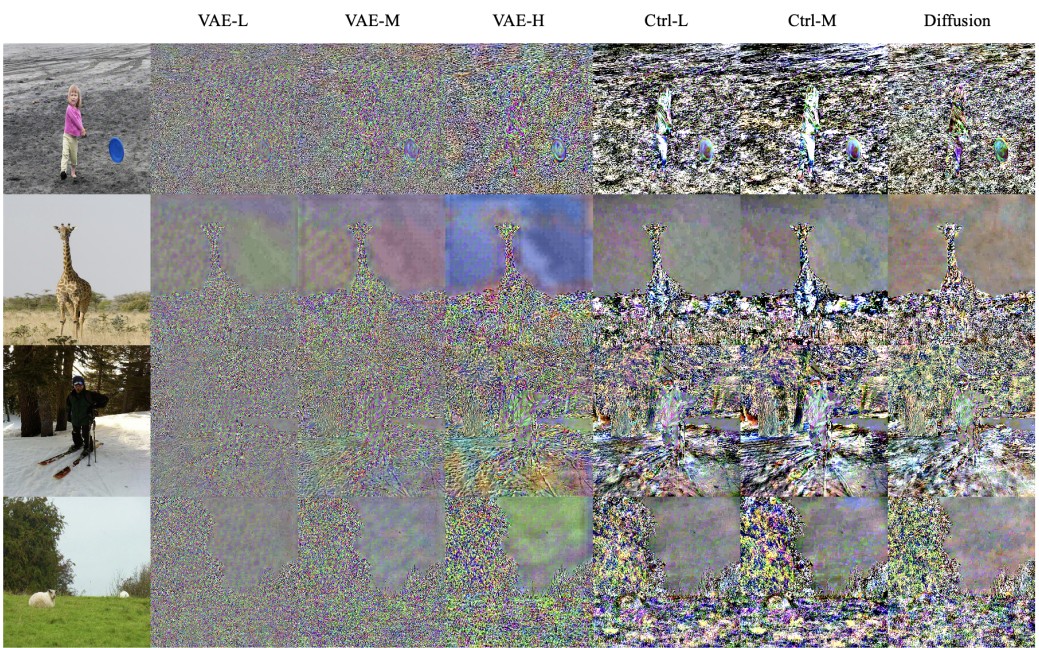

Figure 9: Residuals obtained under different watermark removal methods. The first column shows the watermarked images, while columns 2–7 correspond to the VAE-L, VAE-M, VAE-H, Ctrl-L, Ctrl-M, and Diffusion removal methods, respectively.

According to Table 5, the forgery performance consistently improves as the size of the training dataset increases. When the training set is small (100 samples), the generated images still exhibit high visual quality; however, the forgery effectiveness remains limited, with an average bitwise accuracy of only 0.705 and a success rate of 0.210. This suggests that with insufficient data, the model fails to capture adequate residual information to produce convincing forgeries. As the training set expands (1000–5000 samples), forgery performance improves substantially. Although image quality shows a slight decline compared with the 100-sample setting, it remains largely stable within the range of 1000 to 5000 samples, with no significant further degradation observed.

We further visualize the impact of training sample size on the distribution of forgery bitwise accuracy in Figure 10 . Even with only 100 training samples, most forged samples achieve a bitwise accuracy above 0.6. However, as the evaluation threshold increases, the influence of training size becomes more evident. For instance, at a threshold of 0.60, the proportion of samples exceeding this value is 100% with 5000 training samples and 80% with 100 training samples, indicating only a modest gap. In contrast, at a stricter threshold of 0.95, 70% of samples surpass the threshold with 5000 training samples, whereas only 1% do so with 100 samples, revealing a pronounced disparity.

Table 5: Performance comparison across different training numbers.

|  | AvgBitAcc↑ | Success Rate↑ | PSNR↑ | FID↓ |
|---|---|---|---|---|
| 100 | 0.705 | 0.210 | 39.36 | 5.980 |
| 1000 | 0.894 | 0.819 | 31.71 | 10.76 |
| 3000 | 0.941 | 0.931 | 31.93 | 11.31 |
| 5000 | 0.952 | 0.947 | 31.38 | 13.13 |

## A.5 LIMITATION AND DISCUSSION

Our forgery method is built on the premise that the residuals between watermarked and original images resemble the residuals between watermarked and reconstructed images, this assumption holds well for many post-hoc watermarking schemes that employ image-specific watermark pat-

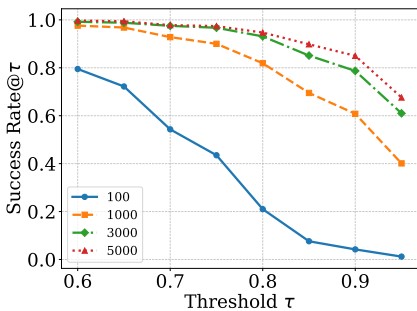

Figure 10: Impact of training sample size on forgery performance.

terns. However, it is less applicable to schemes with fixed watermark patterns. To verify this, we also conducted experiments on such methods.

Table 6: Forgery results of other watermark methods on MS-COCO.

| | Watermark Steganalysis | | | Watermark Faker | | | WForge | | |
|---|---|---|---|---|---|---|---|---|---|
| | Success Rate↑ | PSNR↑ | FID↓ | Success Rate↑ | PSNR↑ | FID↓ | Success Rate ↑ | PSNR↑ | FID↓ |
| Tree-Ring | 1.000 | 20.26 | 26.23 | 0.000 | 18.81 | 59.47 | 0.000 | 29.32 | 17.68 |
| LaWa | 1.000 | 27.23 | 15.45 | 0.000 | 19.97 | 59.85 | 0.502 | 31.64 | 11.05 |
| SleeperMark | 0.910 | 15.23 | 103.7 | 0.000 | 20.23 | 58.93 | 0.000 | 29.57 | 15.51 |
| PRC watermark | 0.000 | 31.68 | 2.384 | 0.000 | 19.57 | 60.01 | 0.000 | 29.10 | 17.66 |

---

**Algorithm 1** WForge

---

**Require:** Watermarked images $\{I_w^{(i)}\}_{i=1}^N$; watermark remover $R'$; forgery model $M_\theta$; norm
             bounds $\epsilon_1, \epsilon_2$
**Ensure:** Forgery operator $F(\cdot)$

 1: **function** RESIDUALCOLLECTION                          ▷ Stage 1: Residual Collection
 2:      **for** $i = 1 \rightarrow N$ **do**
 3:          $I_{\text{rem}}^{(i)} \leftarrow R'(I_w^{(i)})$
 4:          $R_g^{(i)} \leftarrow I_{\text{rem}}^{(i)} - I_w^{(i)}$
 5:          $R_g^{(i)} \leftarrow \Pi_{\epsilon_2}(R_g^{(i)})$
 6:      **end for**
 7:      **return** $\mathcal{T} = \{(I_{\text{rem}}^{(i)}, R_g^{(i)})\}_{i=1}^N$
 8: **end function**

 9: **function** TRAINFORGER($\mathcal{T}$)                        ▷ Stage 2: Train Forgery Model
10:      **while** not converged **do**
11:          **for** each $(I_{\text{rem}}, R_g)$ in $\mathcal{T}$ **do**
12:              $R_p \leftarrow M_\theta(I_{\text{rem}})$
13:              $R_p \leftarrow \Pi_{\epsilon_1}(R_p)$
14:              $\mathcal{L} \leftarrow \|R_p - R_g\|_1$
15:              $\theta \leftarrow \theta - \eta\nabla_\theta\mathcal{L}$
16:          **end for**
17:      **end while**
18:      **return** $M_\theta$
19: **end function**

20: **function** FORGE($I_u$)                            ▷ Stage 3: Forgery on Unwatermarked Images
21:      $R_p \leftarrow M_\theta(I_u)$
22:      $R_p \leftarrow \Pi_{\epsilon_1}(R_p)$
23:      **return** $I_f \leftarrow I_u + R_p$
24: **end function**

25: $\mathcal{T} \leftarrow$ RESIDUALCOLLECTION
26: $M_\theta \leftarrow$ TRAINFORGER($\mathcal{T}$)
27: **return** Forgery operator $F(I_u) =$ FORGE($I_u$)

---

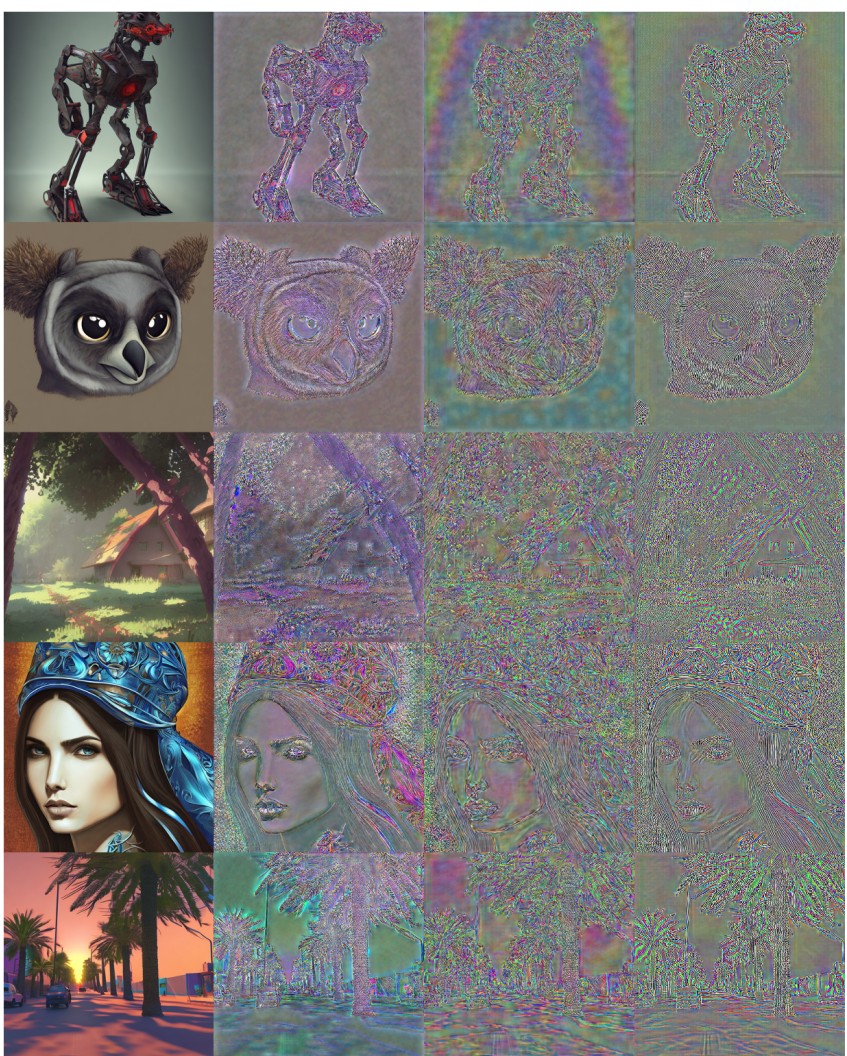

Figure 11: Visualization of original images and three types of residuals (scaled by $\times 10$) in the Stable Signature watermark. Column 1: unwatermarked images $I_u$; Column 2: watermark residuals $r(I_u) = I_w - I_u$; Column 3: residuals $R_g = \mathcal{R}'(I_w) - I_w$ obtained via a VAE-based remover $\mathcal{R}'$; Column 4: our predicted residuals $R_p = M_\theta(I_u)$.

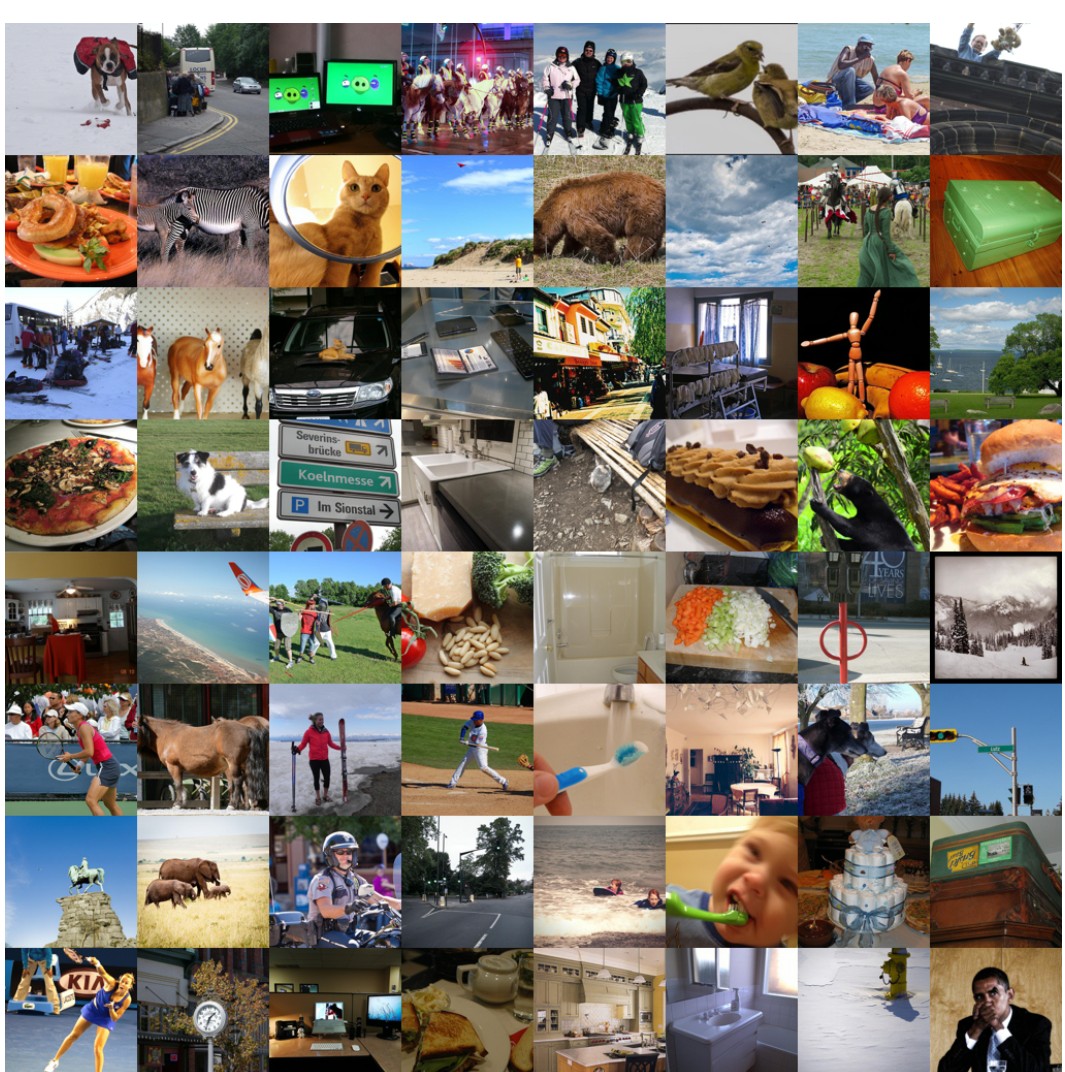

Figure 12: Sample from our forged images of Stable Signature in MS-COCO.

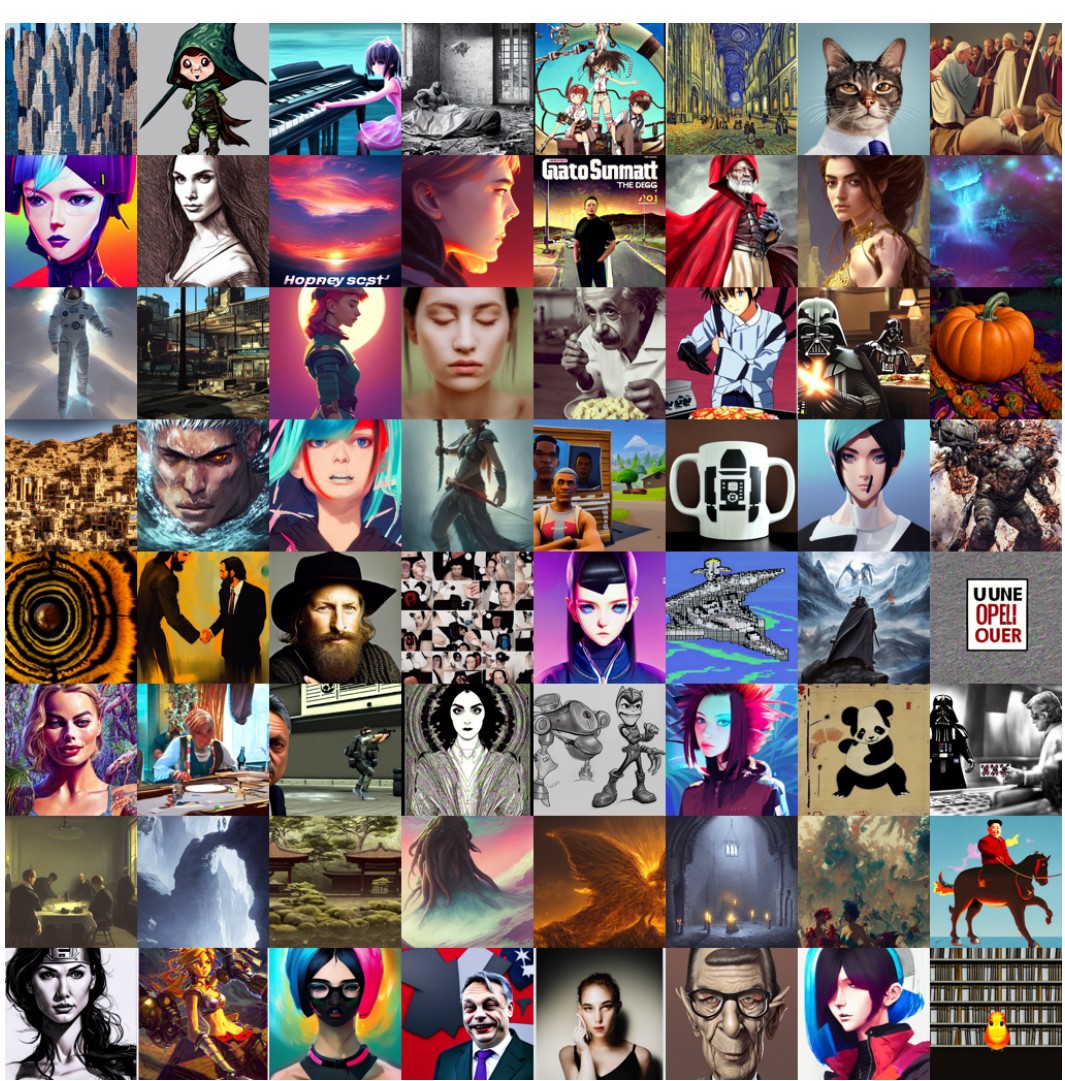

Figure 13: Sample from our forged images of WAM in DiffusionDB.

