# OpenReview forum: "Forging Image Watermarks by Reversing Watermark Removal Attacks"
_ICLR.cc/2026/Conference — Submitted to ICLR 2026_

### Official Review · Reviewer_3rdk · 2025-10-24

**Soundness:** 3
**Presentation:** 4
**Contribution:** 3
**Rating:** 4
**Confidence:** 4

**Summary:**

This paper proposes WForge, a no-box, query-free image watermark forgery method that conceptualizes forgery as the inverse of watermark removal. The core idea is that residual perturbations from watermark removal attacks approximate the underlying watermark signal; by learning these residuals through supervised training, a forger network can add imperceptible perturbations to non-watermarked images so that they are falsely detected as watermarked. Researchers process experiments across multiple datasets, and watermarking methods demonstrate that WForge significantly outperforms prior forgery baselines.

**Strengths:**

1. Good Paper Writing: The paper is exceptionally well-written, featuring a clear structure and a detailed description of the methodology. Furthermore, the authors provide sound theoretical arguments to justify the statistical properties of their detection mechanism.

2. Conceptual Novelty: The straightforward insight that watermark removal and forgery are inverse processes is elegant and leads to a practical, effective attack framework. The authors made extensive experiments on multiple watermarking schemes and datasets to substantiate the method’s generality;

3. Security Relevance: The discovery that removal attacks imply forgery vulnerability is an important and underexplored insight with implications for AI content authenticity frameworks.

**Weaknesses:**

1. Some watermarking techniques used to be forged are outdated, particularly the content-dependent methods, which were developed 5 years ago (e.g., StegaStamp (CVPR 2020) and RivaGAN (arXiv 2019)). It remains uncertain whether the proposed method can effectively counter modern post-processing content-dependent watermarks, such as TrustMark (ICCV 2025) and VINE (ICLR 2025). To ensure a more comprehensive evaluation, experiments involving at least VINE should be included in the experiences. (p.s., I totally understand this may be a lot to run, but some small but convincing experiments would really help. I will consider significantly raising the score if more supportive data about the latest effectiveness can be provided.

2. Potential Concerns about Removal Methods: Since the forger is trained on residuals from a specific remover, its generalization to unseen watermarking-removal pairs may be overstated (e.g., weakness 1 supplementary experiment fails). Besides, the proposed method relies significantly on the availability of effective watermark removal attacks (e.g., VAE-based removers) to generate training residuals. However, if removal is weak or unavailable for certain watermarks, WForge's performance could degrade significantly.

3. Unfinished `Appendix A.4 Observation`: has no contents but only a title.

**Questions:**

Has the author considered combining the results of multiple watermark removal algorithms to create a more comprehensive training dataset? This approach may learn richer prior representations of watermark features, and enhance adaptability when encountering entirely new and unknown watermark paradigms.

Are there any suggestions for the next steps in the development of watermark forging from the perspective of image modality?

---

> ### Author Response · Authors · 2025-11-20
>
> We greatly appreciate your constructive and detailed review. Your precise recommendations have guided us in making focused revisions. Thank you as well for your thoughtful recognition of the paper’s conceptual novelty and security relevance. Your remark that it is “exceptionally well-written” was especially encouraging and truly meant a great deal to us. In the “Questions” section of your review, we greatly appreciated the constructive suggestions and broader perspectives you offered, as well as the sense of responsibility you showed toward the development of this field. Your main points of concern center on the generality of the proposed approach, the influence of the watermark-removal method and layout details. In response to your comments, we have added several additional experiments and provided point-by-point replies to each of your concerns. For the promising directions you suggested, we conducted small-scale experiments, and we also provided some of our personal perspectives on the future development of the field. We would appreciate it if you could reconsider the rating given these revisions. Your insights were very inspiring and meaningful to our work. We look forward to continuing the discussion.

---

> > ### Author Response · Authors · 2025-11-20
> >
> > [1] Shilin Lu, Zihan Zhou, Jiayou Lu, Yuanzhi Zhu, and Adams Wai-Kin Kong. Robust Watermarking Using Generative Priors Against Image Editing: From Benchmarking to Advances. In International Conference on Learning Representations, 2025.
> >
> > [2] Xuandong Zhao, Kexun Zhang, Zihao Su, Saastha Vasan, Ilya Grishchenko, Christopher Kruegel,Giovanni Vigna, Yu-Xiang Wang, and Lei Li. Invisible image watermarks are provably removable using generative ai. In Advances in neural information processing systems, 2024.
> >
> > [3] Yepeng Liu, Yiren Song, Hai Ci, Yu Zhang, Haofan Wang, Mike Zheng Shou, and Yuheng Bu. Image watermarks are removable using controllable regeneration from clean noise. In International Conference on Learning Representations, 2025.
> >
> > [4] Tom Sander, Pierre Fernandez, Alain Oliviero Durmus, Teddy Furon, and Matthijs Douze. Watermark anything with localized messages. In International Conference on Learning Representations, 2025.

---

> ### Author Response · Authors · 2025-11-20
>
> Comment#1: More Watermarking Methods
>
> Answer#1:
> VINE[1] is currently one of the most advanced and highly robust watermarking approaches, representing the latest and most resilient progress in the field of watermarking. Thank you for the suggestion—we agree that it is necessary to include this method. We will cite VINE accordingly and add the corresponding experiments.
> Our article focuses on the conclusion that the existence of a successful removal attack for an image watermarking method implies that the watermark can also be forged correspondingly.  VINE's design has already taken robustness against existing image regeneration techniques into account, which means that there is currently no highly effective removal method for it. We plan to first evaluate how different removal methods perform on VINE, in order to identify the most effective possible removal strategy for our experiments.
> Experiment1: Removal Performance in VINE
> We generated 5,000 VINE-R watermarked images using the MSCOCO training dataset, and then conducted experiments with the following five watermark-removal methods. We evaluated both the effectiveness and utility of each removal approach, and compared the quality of the watermarked images before and after removal.
>
> | Remove Method | Original AvgBitAcc | After-Removal AvgBitAcc↓ | PSNR↑ | FID↓  |
> | ------------- | ------------------ | ------------------------- | ----- | ----- |
> | VAE-L         | 1.000              | 0.999                     | 33.01 | 7.095 |
> | VAE-M         | 1.000              | 0.993                     | 29.85 | 16.98 |
> | VAE-H         | 1.000              | 0.941                     | 27.14 | 36.76 |
> | Ctrl-L        | 1.000              | 0.842                     | 24.02 | 4.826 |
> | Ctrl-M        | 1.000              | 0.722                     | 20.80 | 6.979 |
>
> We observe that VINE indeed demonstrates stronger robustness under various settings (for example, its watermark is almost impossible to remove even under VAE-based attacks[2] of different intensities). However, the latest watermark removal method, CtrlRegen[3], can partially remove the watermark to some extent.
>
> Due to time constraints, we selected three high-performing watermark removal methods as “successful removal attacks” to train the Forger Network (VAE FT-Decoder). Specifically, for each remover, we applied WForge to convert the removal process into a forgery process on the 5,000 watermark-removed images, and trained the model for 50 epochs. We then evaluated the model on 100 images from the MSCOCO test dataset. The resulting forged outputs are shown below:
>
> |        | Forgery AvgBitAcc↑ | PSNR↑ | FID↓  |
> | ------ | ------------------ | ----- | ----- |
> | VAE-H  | 0.590              | 26.36 | 39.06 |
> | Ctrl-L | 0.712              | 29.40 | 11.70 |
> | Ctrl-M | 0.610              | 27.01 | 17.81 |
>
>
> Furthermore, for Ctrl-L, we conducted additional evaluations on a larger test set and reported more comprehensive experimental results, as shown below:
>
> | Dataset     | Forgery AvgBitAcc↑ | PSNR↑ | FID↓  |
> | ----------- | ------------------ | ----- | ----- |
> | MSCOCO      | 0.712              | 29.40 | 11.70 |
> | ImageNet    | 0.698              | 28.52 | 12.64 |
> | DiffusionDB | 0.702              | 29.08 | 12.07 |
>
>
> For VINE, WForge achieves an AvgBitAcc of approximately 0.7, indicating that it can partially realize a no-box forgery attack on this method. Although the performance is not as strong as that observed on other watermarking schemes, considering that VINE represents the most recent and robust watermarking approach, we regard this result as acceptable. Moreover, it demonstrates the potential of leveraging the reversal of more effective watermark removal procedures to enable forgery attacks.
>
> In fact, our approach should not be interpreted as lacking generalization to modern post-processing, content-dependent watermarking methods. All of our experiments are conducted on WAM(ICLR 2025)[4]. The misunderstanding likely comes from the fact that WForge is not designed to be universally applicable to all watermark types. Instead, it specifically targets content-dependent watermarking methods. This is because content-agnostic watermarks have already been shown to be highly vulnerable, as their fixed patterns can be forged simply through averaging. In contrast, content-dependent watermarks have long been considered difficult to forge, and demonstrating that even this class of watermarks is insecure is precisely the core of our contribution. We believe this is what gives our work its significance.

---

> ### Author Response · Authors · 2025-11-20
>
> Comment#2: Dependence on Watermark-Removal Effectiveness
>
> Answer#2: In this section, we jointly address Weaknesses 2 and Questions 1.
>
> > Since the forger is trained on residuals from a specific remover, its generalization to unseen watermarking-removal pairs may be overstated.
>
> The comment suggests that the generalization of our method to unseen watermarking–removal pairs may have been overstated. We understand this concern as follows: for a given watermarking scheme, WForge currently relies on a single remover to obtain the watermarking–removal pairs, and the forger therefore learns residual patterns produced by that specific remover. A more principled approach would be to incorporate multiple removers and derive a more generalized representation of watermark residuals through appropriate processing. This line of reasoning naturally connects to the concrete suggestion raised in Question 1—namely, constructing a broader dataset using multiple removers. This is truly an insightful observation, and we greatly appreciate the constructive suggestion built upon our original design.
> Building on this insight, we conducted a small-scale preliminary experiment. Due to time constraints, the experimental setup is as follows:
> We use VAE fine-tuning as the forge network, trained for 40 epochs, and 100 ImageNet images as the test set. We constructed a remover pool consisting of five watermark-removal methods: VAE-M[2], VAE-L[2], VAE-H[2], Diffusion[2], and Ctrl-M[3]. For training dataset, we selected 1,000 WAM-watermarked images and built it using the following four different approaches:
>
> - Method1(VAE-M): All 1,000 watermarked images were removed using VAE-M.
> - Method2(Mixed): 200 images each were removed by VAE-M, VAE-L, VAE-H, Diffusion, and Ctrl-M.
> - Method3(Average): For 1,000 images, we computed the average residual using results from VAE-M, VAE-L, VAE-H, Diffusion, and Ctrl-M.
>
> |             | AvgAcc↑ | PSNR↑ | FID↓  |
> |-------------|---------|-------|-------|
> | VAE-M  | 0.810   | 32.70 | 12.65 |
> | Mixed       | 0.516   | 26.49 | 34.10 |
> | Average     | 0.821   | 30.68 | 12.90 |
>
> Mixed yields the weakest results. We conjecture that this is because, when combining removed images produced by multiple removers, some removers generate low-quality removals that substantially degrade the overall quality of the training data. The performance of Average is close to that of our current method: it achieves a slightly higher forgery success rate but produces images of slightly lower quality. However, its computational and operational cost is substantially higher than that of the original approach. Therefore, we consider it reasonable to adopt the simpler VAE-M setting for validating our conclusions. Nonetheless, we plan to explore more efficient or effective alternatives in future work.
>
> > If removal is weak or unavailable for certain watermarks, WForge's performance could degrade significantly.
>
> Our conclusion is that the existence of a successful removal attack for an image watermarking method implies that the watermark can also be forged correspondingly. We seek to demonstrate to the community a promising and novel methodology for building no-box forgery attacks, instead of focusing on the removal-attack domain itself. The primary objective of the field of watermark-removal attacks is to develop a successful removal attack as we need. As the field continues to evolve rapidly, we anticipate that more powerful techniques will continue to emerge. Such advances will, in turn, further strengthen and enhance the effectiveness of our proposed forgery attack.
>
> -----
>
> Comment#3: Unfinished Part
>
> Answer#3: Thank you for pointing this out. We have thoroughly reorganized the appendix, and this issue should now be resolved.

---

> ### Author Response · Authors · 2025-11-20
>
> Comment#4: Development of Watermark Forgery
>
> Answer#4:
>
> Forge attacks have long been an open challenge in watermarking robustness research, and have been widely considered infeasible in the no-box setting. We argue that approaching the problem from a residual perspective offers a new and promising direction: reframing watermark forgery as the task of recovering effective watermark residuals. This perspective opens a new avenue for no-box forgery, and we look forward to more high-quality work emerging along this line. During our investigation, we developed several related insights.
>
> **Does strong watermark robustness inadvertently make watermark forgery more feasible?**
>
> Intuitively, a robust watermark must preserve decodability under heavy distortions such as noise, compression, and cropping. This implies that its internal watermark pattern must be sufficiently salient, stable, and often consistent across different contents. While this design improves robustness, it also introduces a critical risk: the more robust the watermark, the more likely its residual signal becomes a content-agnostic fixed pattern, which can be extracted and forged by an attacker.
> We provide two pieces of evidence supporting this hypothesis:
> Many recently proposed robust watermarking methods can be forged simply by averaging a small number of watermarked samples, which reveals a clear, stable fixed watermark residual like Lawa(ECCV 2024), Tree-Ring(NeurIPS 2023), SleeperMark(CVPR 2025).
>
>
> In our experiments with StegaStamp (CVPR 2020), we observed that adversarial robustness training further pushes the watermark residual toward a fixed pattern. Although the intention is to enhance robustness against distortions, from an attacker’s perspective this process forces the model to embed a more uniform and predictable residual across different images.
> Viewed from a forgery perspective, this leads to an interesting conclusion: many “robust” watermarking methods may, in practice, be learning a global fixed residual template through complex architectures and training strategies, and then injecting this template into all content. This phenomenon becomes especially visible under forgery testing—what appears to be a content-adaptive robust watermark may, in reality, behave like a universal, easily forgeable pattern.
> This structural tension between robustness and forgery resistance highlights a deeper weakness in current watermark design, and we believe it represents a promising direction for future research.
>
> **Incorporating Forgery Attacks into Watermark Robustness Evaluation**
>
> Compared with removal attacks, which are commonly studied in watermark robustness research, the community has paid far less attention to forgery attacks. Watermark Steganalysis focuses on average-based forgeries derived from global statistical patterns, whereas our proposed WForge targets the forgery of individual watermark patterns. These two attack strategies—one global, one instance-specific—are complementary. We strongly recommend that both types of forgery attacks be included in any comprehensive watermark robustness evaluation.
>
> Our experiments show that major existing watermarking methods, including Lawa(ECCV 2024), Tree-Ring(NeurIPS 2023), SleeperMark(CVPR 2025), StegaStamp (CVPR 2020), WAM(ICLR 2025), RivaGAN (arXiv 2019) and Stable Signature(ICCV 2023) can be totally forged either by Watermark Steganalysis or by WForge. This finding poses a serious challenge to the robustness of current watermark designs. Successful forgery fundamentally undermines the ownership verification capability of watermarks: once an attacker can synthesize a “valid” watermark, the original claim of ownership collapses, effectively nullifying the purpose of watermarking. We encourage future watermarking approaches to consider not only robustness to remove attacks but also explicit resistance to forgery.
>
> In an era where AI-generated fake images and videos are increasingly widespread, we hope that this work raises awareness of the importance of forgery attacks. Our goal is to motivate the community to develop more reliable and proactive detection mechanisms, and to inspire the design of watermarking methods that are truly robust—even in the presence of advanced forgery attacks.

---

> > ### Comment · Reviewer_3rdk · 2025-11-27
> >
> > Thank you to the author for thoroughly addressing my concerns. Your research offers valuable insights into leveraging existing mature methods (i.e., the removers) and extending their application to the domain of watermark forging.
> >
> > I truly appreciate your additional experimental analysis and the clear explanations you provided. In line with my previous commitment, I will increase my rating accordingly.

---

> > > ### Author Response · Authors · 2025-11-27
> > >
> > > We truly appreciate your high recognition of our work and the increased score. Your insightful comments have significantly helped us enhance the revised manuscript. Thank you again for the time and effort you dedicated to the review!

---

### Official Review · Reviewer_FcRK · 2025-10-28

**Soundness:** 3
**Presentation:** 2
**Contribution:** 2
**Rating:** 4
**Confidence:** 2

**Summary:**

This paper studies how to perform watermark forgery attacks in a no-box and query-free setting by leveraging reverse watermark removal attacks. Concretely, the authors investigate whether one can reverse an existing watermark-removal method and use that to train a forger network that produces forged watermarks without access to the original watermarking pipeline or queries to it.

**Strengths:**

1. The problem of forging watermarks in a no-box, query-free setting is realistic and the threat model is reasonable—practical and worthy of study.

2. The proposed approach is well-motivated. Empirical results indicate that the method can work under the evaluated settings, supporting the paper’s claims of feasibility.

**Weaknesses:**

1. The paper lacks a main flowchart or an algorithm diagram in Section 4 that clearly lays out the end-to-end pipeline. A visual summary or pseudocode would greatly improve clarity.

2. The approach appears heavily dependent on the effectiveness of the underlying watermark-removal method. If the removal step fails or is weak, subsequent training of the forger network may be unreliable. The manuscript does not sufficiently explain how the reliability of the removal stage is ensured, nor quantify how removal quality affects downstream forgery. This raises concerns about strong implicit assumptions on the attacker’s knowledge or capabilities.

3. The experimental set of watermarking methods is limited, which weakens claims about generality. It is unclear whether the approach generalizes across diverse watermark schemes (e.g., FNNS, TrustMark, and other modern methods).

4. The paper does not investigate robustness to unknown watermark parameters (e.g., strength, spatial placement, embedding hyperparameters). It is unclear whether the method can forge or remove watermarks when these parameters vary or are unknown.

**Questions:**

Does the method require any implicit or explicit knowledge about the watermark (e.g., type, strength, embedding scheme) to succeed? If so, please clarify these assumptions and discuss their realism in your threat model.

---

> ### Author Response · Authors · 2025-11-20
>
> Thank you for your thoughtful review. We appreciate your acknowledgment of the practical value of our problem setting and the sound motivation behind our approach. Your feedback is highly constructive and greatly appreciated. Your main concerns pertain to the article’s clarity, the influence of the watermark-removal method, the generality of the proposed approach, and its robustness to unknown watermark parameters. In response to your comments, as we fully agree that such a visualization is important for clearly conveying our method, we have added an algorithm diagram in appendix to clarify the overall workflow. Furthermore, following your suggestions, we conducted additional experiments and provided corresponding explanations, as detailed below. Please feel free to share any further concerns—we would be more than willing to address them during the discussion. We kindly ask you to consider the possibility of revisiting the rating in light of these improvements.
>
> ------
>
> Comment#1: Article’s Clarity
>
> Answer#1: This is a very valuable suggestion. Our `Section 4 (Methodology)` indeed lacked an algorithm diagram to summarize the entire approach. Due to space limitations, the details can be found in appendix in the updated version of the paper.
>
> -------
> Comment#2: Dependence on Watermark-Removal Effectiveness
>
> Answer#2:
>
> **2.1. How the reliability of the removal stage is ensured**
>
> Our conclusion is that the existence of a successful removal attack for an image watermarking method implies that the watermark can also be forged correspondingly. We seek to demonstrate to the community a promising and novel methodology for building no-box forgery attacks, instead of focusing on the removal-attack domain itself. The primary objective of the field of watermark-removal attacks is to develop a successful removal attack as we need. As the field continues to evolve rapidly, we anticipate that more powerful techniques will continue to emerge. Such advances will, in turn, further strengthen and enhance the effectiveness of our proposed forgery attack.
>
> **2.2. Quantify how removal quality affects downstream forgery**
>
>
> In `Section 5.3 (Ablation Study)`, we quantified how removal quality affects downstream forgery. To comprehensively evaluate the impact of removal quality, we adopted different removal methods (VAE remove[1], diffusion remove[1], Ctrl remove[2]) as well as different removal strengths (Low, Medium, High). The corresponding visualizations can be found in `Figure 4`, and the related analysis is presented in `WForge using different removal methods`. This comment may be raised because the focus of this section was not sufficiently emphasized. If necessary, we plan to restructure this part to better highlight the central theme of removal quality.
>
> -----
>
> Comment#3: More Watermarking Methods
>
> Answer#3: We have now added experiments on VINE[3], as detailed in our response to `Reviewer 3 (Answer #1)`. These results show that our forgery approach also applies to this latest watermarking method. Following your suggestion, we are running experiments on TrustMark[4] and FNNS[5] as well. We are eager to include these results.

---

> ### Author Response · Authors · 2025-11-20
>
> Comment#4: Robustness to unknown watermark parameters
>
> Answer#4:
> Weaknesses 4 and Question appear to refer to the same issue, and we think there might be some misunderstanding on this point.
> WForge is designed for the scenario where watermark parameters vary or are completely unknown. In fact, WForge does not require any access to watermark parameters at all—neither implicitly nor explicitly. As described in `Section 3.1 (Threat Model)`, WForge operates in a strict no-box setting, where the attacker does not know: watermark strength, spatial placement, embedding hyperparameters, watermark type or embedding scheme, or any internal details of the algorithm.The only assumptions are: (i) there exists a watermark detector, and (ii) the input images are watermarked according to that detector.
> We believe this misunderstanding stems from an insufficiently clear explanation of our method. To address this, we have added an algorithm diagram to better illustrate the workflow of WForge in appendix. As shown in the updated version, WForge relies solely on watermarked images to perform forgery, without requiring any knowledge of the watermarking parameters or embedding process. It naturally handles unknown or varying watermark parameters, and does not rely on any consistent or fixed embedding configuration. Thus, the concern raised in the comment does not apply to our threat model.
>
>
> [1] Xuandong Zhao, Kexun Zhang, Zihao Su, Saastha Vasan, Ilya Grishchenko, Christopher Kruegel,Giovanni Vigna, Yu-Xiang Wang, and Lei Li. Invisible image watermarks are provably removable using generative ai. In Advances in neural information processing systems, 2024.
>
> [2] Yepeng Liu, Yiren Song, Hai Ci, Yu Zhang, Haofan Wang, Mike Zheng Shou, and Yuheng Bu. Image watermarks are removable using controllable regeneration from clean noise. In International Conference on Learning Representations, 2025.
>
> [3] Shilin Lu, Zihan Zhou, Jiayou Lu, Yuanzhi Zhu, and Adams Wai-Kin Kong. Robust Watermarking Using Generative Priors Against Image Editing: From Benchmarking to Advances. In International Conference on Learning Representations, 2025.
>
> [4] Bui, Tu and Agarwal, Shruti and Collomosse, John. TrustMark: Robust Watermarking and Watermark Removal for Arbitrary Resolution Images.In IEEE International Conference on Computer Vision, 2025.
>
> [5] Kishore, Varsha and Chen, Xiangyu and Wang, Yan and Li, Boyi and Weinberger, Kilian Q. Fixed neural network steganography: Train the images, not the network.  In International Conference on Learning Representations, 2021

---

> > ### Comment · Reviewer_FcRK · 2025-11-27
> >
> > Thank you for your response. However, I still have some concerns.
> >
> > 1. The method relies on a forgery network learning residual patterns, but the manuscript still lacks a clear framework diagram. This omission makes it difficult to understand how WForge operates end-to-end.
> >
> > 2. The authors’ rebuttal suggests that a successful removal attack is a prerequisite for effective forgery. This is a very strong and often unrealistic assumption. If the watermark is not fully removed, can the forged watermark still succeed? In what real-world scenarios can this assumption reliably hold?
> >
> > 3. The authors state that no watermark parameters are needed and that the only requirement is a watermark detector. However, trained detectors may implicitly encode watermark strength, embedding position, or hyperparameters. Does relying on such a detector effectively mean having access to these parameters?

---

> > > ### Author Response · Authors · 2025-11-29
> > >
> > > Thank you for the follow-up comments. Your insightful feedback has greatly helped us improve the revised manuscript. Please see our response to your concerns below.
> > >
> > > ---
> > >
> > > 1. In the revised version, we have included a clear and easy-to-understand framework figure in the main text (Figure 1). We have also added an algorithm diagram in the Appendix (Algorithm 1) to better illustrate the workflow of WForge. We believe these additions will help readers understand how WForge operates end-to-end.
> > >
> > > ---
> > >
> > > 2. There might be a misunderstanding. We do not suggest that a successful removal attack is a necessary prerequisite for effective forgery. In the no-box setting, even if the watermark is not fully removed, WForge can still partially forge the watermark. For example, when using Ctrl-L as the remover for the VINE [1] watermarking method, we obtain the following results:
> > >
> > > | Remove Method | Original AvgBitAcc | After-Removal AvgBitAcc↓ | PSNR↑ | FID↓  |
> > > | ------------- | ------------------ | ------------------------- | ----- | ----- |
> > > | Ctrl-L        | 1.000              | 0.842                     | 24.02 | 4.826 |
> > >
> > > The After-Removal AvgBitAcc is 0.842, indicating that the watermark is only slightly removed by Ctrl-L, as a complete removal would result in an AvgBitAcc of approximately 0.5. Nevertheless, we find that WForge is still capable of forging the watermark. The results are as follows:
> > >
> > > | Dataset     | Forgery AvgBitAcc↑ | PSNR↑ | FID↓  |
> > > | ----------- | ------------------ | ----- | ----- |
> > > | MSCOCO      | 0.712              | 29.40 | 11.70 |
> > > | ImageNet    | 0.698              | 28.52 | 12.64 |
> > > | DiffusionDB | 0.702              | 29.08 | 12.07 |
> > >
> > > Moreover, if the attacker is more capable (consistent with the setting in previous work WatermarkFaker[2]), i.e., able to collect both watermarked images and their corresponding unwatermarked versions, they can directly replace the watermark-removed outputs with the clean images, making the training of the forgery network even easier and more effective.
> > >
> > > On the other hand, the assumption that an effective watermark removal method exists for a given watermarking method is neither strong nor unrealistic. Several prior works have demonstrated successful watermark removal attacks against many watermarking methods in the no-box setting. For example, Steganalysis [3] has been shown to effectively remove state-of-the-art content-agnostic watermarking methods such as TreeRing (NeurIPS 2023) and GaussianShading (CVPR 2024). Zhao et al. [4] reported that no-box regeneration attacks can remove StegaStamp (CVPR 2020) watermarks. In addition, WAVES [5] demonstrated that state-of-the-art image watermarking methods, including TreeRing, StegaStamp, and Stable Signature (ICCV 2023), are also not robust against watermark removal attacks.
> > >
> > > ---
> > >
> > > 3. We apologize for not making this point clear enough in the previous rebuttal, which may have led to a misunderstanding. What we meant is that the detector is used only for evaluation—that is, to assess whether the forgery attack is successful—but it is not required during the forgery process. WForge operates entirely in a no-box setting and requires only a set of watermarked images, which is highly practical in real-world scenarios. For example, an attacker can collect watermarked images directly from a GenAI service that embeds watermarks into its outputs. Beyond these watermarked images, no additional information is needed for our forgery attack, including any watermark detector or watermarking parameters.
> > >
> > > ---
> > >
> > > We hope these clarifications and added experimental results and figure/algorithm address your concerns.
> > >
> > > ---
> > >
> > > [1] Shilin Lu, Zihan Zhou, Jiayou Lu, Yuanzhi Zhu, and Adams Wai-Kin Kong. Robust watermarking using generative priors against image editing: from benchmarking to advances. ICLR, 2025.
> > >
> > > [2] Ruowei Wang, Chenguo Lin, Qijun Zhao, and Feiyu Zhu. Watermark faker: Towards forgery of digital image watermarking. ICME, 2021.
> > >
> > > [3] Pei Yang, Hai Ci, Yiren Song, and Mike Zheng Shou. Can simple averaging defeat modern watermarks? NeurIPS, 2024.
> > >
> > > [4] Xuandong Zhao, Kexun Zhang, Zihao Su, Saastha Vasan, Ilya Grishchenko, Christopher Kruegel,Giovanni Vigna, Yu-Xiang Wang, and Lei Li. Invisible image watermarks are provably removable using generative ai. NeurIPS, 2024.
> > >
> > > [5] An, Bang and Ding, Mucong and Rabbani, Tahseen and Agrawal, Aakriti and Xu, Yuancheng and Deng, Chenghao and Zhu, Sicheng and Mohamed, Abdirisak and Wen, Yuxin and Goldstein, Tom and Huang, Furong. WAVES: Benchmarking the Robustness of Image Watermarks. ICML, 2024.

---

### Official Review · Reviewer_avHU · 2025-10-31

**Soundness:** 3
**Presentation:** 3
**Contribution:** 3
**Rating:** 6
**Confidence:** 5

**Summary:**

This paper proposes WForge, a method for forging content-dependent image watermarks. WForge first leverages a generative watermark-removal model to obtain a clean image from a watermarked one, and then extracts the watermark residual by taking their difference. It subsequently trains a network on large-scale data to predict watermark residuals for arbitrary images, thereby generalizing the residual estimation process. By adding the predicted residuals back to clean images—following watermark steganalysis principles—the method successfully forges content-dependent watermarks. Experimental results show that WForge achieves higher attack success rates than the baseline methods Steganalysis and Watermark Faker on content-dependent watermarking schemes.

**Strengths:**

1. The paper is interesting and technically insightful. The idea of obtaining watermark residuals through a generative removal process and then learning a generalized residual predictor is creative. The authors effectively extend watermark steganalysis techniques to the more challenging setting of content-dependent watermarks.

2. Extensive ablation studies are conducted to validate the effectiveness of the proposed method.

**Weaknesses:**

1. The optimal configuration (i.e., choice of watermark removal method and residual predictor model) appears to vary across different watermarking algorithms. In real-world scenarios, since the watermark decoder is inaccessible, the forgery performance is uncertain, making it difficult to determine an appropriate configuration in practice.

**Questions:**

For the Success Rate metric, it is unclear why the authors chose the threshold value τ = 0.8. Typically, τ is selected based on the False Positive Rate (FPR). What is the corresponding FPR for this setting?

---

> ### Author Response · Authors · 2025-11-20
>
> We greatly appreciate the time and care you put into reviewing our work. Your encouraging feedback and your recognition of the novelty and technical insight of our work mean a great deal to us. Regarding your primary concerns about the Success Rate metric and the optimal configuration, we address these two aspects separately. If any further issues arise, we would be happy to address them in the discussion. We also hope the improvements we made can be taken into consideration.
>
> ------
>
> Comment#1: Success Rate Metric
>
> Answer#1:
> To determine whether an image contains the watermark produced by the given method, the watermark detector employs the bitwise accuracy (BA) of the decoded watermark as the decision criterion. Given an image $I$, the decoder produces a watermark $D(I)$. The image is deemed watermarked if the bitwise accuracy relative to the ground-truth watermark $w$ exceeds the threshold $\tau$ :
>
> $
> \mathrm{BA}(D(I), w) > \tau.
> $
>
> We follow the single-tail detector formulation proposed in WEvade[1]. Let the ground-truth watermark $w$ be an n-bit string. For any original image $I_o$ that does not contain the watermark, the decoded watermark $D(I_o)$ is statistically independent of $w$, since the service provider samples $w$ uniformly at random from all possible n-bit sequences. Consequently, each bit of $D(I_o)$ matches the corresponding bit of $w$ with probability 0.5, and the total number of matching bits $m$ follows a binomial distribution:
>
> $
> m \sim B(n, 0.5).
> $
>
> Based on this property, the probability that an original image is incorrectly classified as watermarked (i.e., the false positive rate of the watermark detector) is the probability that the bitwise accuracy exceeds the threshold $\tau$:
>
> $
> \mathrm{FPR_s}(\tau) = \Pr\left(\mathrm{BA}(D(I_o), w) > \tau\right) = \Pr(m > n\tau) = \sum_{k=\lceil n\tau\rceil}^{n} \binom{n}{k} 2^{-n}.
> $
>
> To ensure that the false positive rate does not exceed a prescribed bound $\eta$
> (e.g., $\eta = 10^{-4}$), the threshold $\tau$ must be chosen such that:
>
> $
> \mathrm{FPR_s}(\tau) < \eta.
> $
>
> In our main experiment, the watermark
> length is set to $n = 32$, and the detection threshold is chosen as $\tau = 0.8$. Under this
> setting, the false positive rate (FPR) of the detector can be computed as follows.
>
> $
> n\tau = 32 \times 0.8 = 25.6, \qquad k_0 = \lceil n\tau \rceil = 26.
> $
>
> $
> \mathrm{FPR_s}(0.8) = \Pr(m \ge 26) = \sum_{k=26}^{32} \binom{32}{k} 2^{-32}\approx 2.68 \times 10^{-4}.
> $
>
> Therefore, with $n = 32$ and $\tau = 0.8$, the detector produces a false positive rate of approximately $2.68 \times 10^{-4}$.
> We choose $\tau = 0.8$ as the default threshold, as it provides the most intuitive and representative effectiveness for readers. Moreover, `Figure 2` presents a comprehensive evaluation of the results under different values of $\tau$. Therefore, we think the choice of $\tau = 0.8$ should not be a significant concern.

---

> ### Author Response · Authors · 2025-11-20
>
> Comment#2: Optimal Configuration
>
> Answer#2:
>
> Optimal Configurations Are Hard, Effective Ones Are Not: This is an insightful and interesting observation, and you are partially correct. In real-world scenarios (no-box setting in our paper as well), since the watermark decoder is inaccessible, determining the optimal forgery configurations is difficult, but we are still able to find relatively appropriate configurations.
>
> About Experimental Design: This perspective is closely related to experimental design in some way. In fact, we had considered this issue when designing our main experiments. Specifically, we deliberately adopted exactly the same configuration across multiple watermarking schemes (RivaGAN[2], StegaStamp[3], Stable Signature[4], and WAM[5]) to demonstrate that, under a simple default setting, WForge can effectively forge these. The results in `Table 1`  also support this point: even under this unified default configuration, the forgery performance remains consistently strong and we further provide detailed explanations and additional analyses in the `Appendix A.2.3`.
>
> Determining the optimal configuration for each watermarking method under the current no-box setting is inherently a difficult problem. The amount of information available in this setting is extremely limited—most critically, we cannot even access the watermark decoder. This makes it fundamentally infeasible to identify a per-method optimal configuration.
>
> We welcome any additional suggestions and would be glad to continue the discussion.
>
>
> [1] Zhengyuan Jiang, Jinghuai Zhang, and Neil Zhenqiang Gong. Evading watermark based detection of ai-generated content. In Proceedings of the 2023 ACM SIGSAC Conference on Computer and Communications Security, 2023.
>
> [2] Kevin Alex Zhang, Lei Xu, Alfredo Cuesta-Infante, and Kalyan Veeramachaneni. Robust invisible video watermarking with attention. arXiv preprint arXiv:1909.01285, 2019.
>
> [3] Matthew Tancik, Ben Mildenhall, and Ren Ng. Stegastamp: Invisible hyperlinks in physical photographs. In Proceedings of the IEEE/CVF conference on computer vision and pattern recognition, 2020.
>
> [4] Pierre Fernandez, Guillaume Couairon, Herve Jegou, Matthijs Douze, and Teddy Furon. The stable signature: Rooting watermarks in latent diffusion models. In Proceedings of the IEEE/CVF International Conference on Computer Vision, 2023
>
> [5] Tom Sander, Pierre Fernandez, Alain Oliviero Durmus, Teddy Furon, and Matthijs Douze. Watermark anything with localized messages. In International Conference on Learning Representations, 2025.

---

> > ### Comment · Reviewer_avHU · 2025-11-27
> >
> > Thanks for the authors' response. I have no further concerns.

---

> > > ### Author Response · Authors · 2025-11-27
> > >
> > > Thank you for the reply. We are happy to see that we have addressed your concerns!
> > >
> > > Your comments have been invaluable in helping us strengthen the revised version of our work. If you have no further concerns, we would appreciate your consideration in raising the score.

---

### Comment · Area_Chair_sdrM · 2025-11-28
**The Author/Reviewer Discussion Phase deadline is approaching**

Dear Reviewers,

The Author/Reviewer Discussion Phase deadline is approaching. If you have not responded to authors’ rebuttal, please read and give your feedback asap.

Regards,
AC

---

### Meta-Review · Area_Chair_t8Hr · 2026-01-05

**Summary:**

Proposes WForge, a no-box, query-free watermark forgery attack. Key idea: watermark removal residuals approximate watermark signals; train a forger network on (watermarked ->removed) residuals, then add predicted residuals to benign images so they are falsely detected as watermarked. Evaluated on multiple datasets and watermark methods; claims strong gains over forgery baselines and argues “successful removal implies feasible forgery”.

**Reviewer Concerns:**

Strengths (supported by reviews)

Important and realistic threat model: forgery in a no-box, query-free setting is practically relevant (FcRK) and underexplored.

Clean conceptual insight: “forgery as inverse of removal” is elegant and security-relevant (3rdk).

Solid empirical package + ablations: multiple datasets/methods; added experiments on VINE (ICLR’25) and clarified metrics/figures; reviewers avHU and 3rdk explicitly said concerns were resolved.

Main remaining concern: Dependence on removers / realism of assumption (FcRK)
Concern that forgery effectiveness depends on having a sufficiently strong removal method. Although authors clarified the detector is used only for evaluation and showed that partial removal can still yield nontrivial forgery, the reviewer remained unconvinced and kept a low-confidence assessment.

**Reviewer Scores:**

Reviewer avHU: Already engaged fully in discussion; concerns were addressed and the final score (6) appropriately reflects their assessment.

Reviewer FcRK: Participated but remained cautious; even with fuller discussion, the score would likely remain at 4, reflecting continued concerns about realism and dependence on removal strength.

Reviewer 3rdk: Engaged in discussion and acknowledged that concerns were largely addressed, but still submitted a final score of 4; with full participation reflected strictly in the score, it would likely remain 4, though their comments indicate a borderline or positive stance.

---

### Decision · Program_Chairs · 2026-01-26

Reject